# In silico prediction of ARB resistance: A first step in creating personalized ARB therapy

**Shane D. Anderson, Asna Tabassum**, **Jae Kyung Yeon**, **Garima Sharma, Priscilla Santos**, **Tik Hang Soong**, **Yin Win Thu, Isaac Nies**, **Tomomi Kurita, Andrew Chandler**, **Abdelaziz Alsamarah, Rhye-Samuel Kanassatega**, **Yun L. Luo**\*, **Wesley M. Botello-Smith**\*, **Bradley T. Andresen**\*

Department of Pharmaceutical Sciences, College of Pharmacy, Western University of Health Sciences, Pomona, California, United States of America

\* luoy@westernu.edu (YLL); wbotellosmith@westernu.edu (WMB-S); bandresen@westernu.edu (BTA)

## Abstract

Angiotensin II type 1 receptor (AT$_1$R) blockers (ARBs) are among the most prescribed drugs. However, ARB effectiveness varies widely, which may be due to non-synonymous single nucleotide polymorphisms (nsSNPs) within the AT$_1$R gene. The AT$_1$R coding sequence contains over 100 nsSNPs; therefore, this study embarked on determining which nsSNPs may abrogate the binding of selective ARBs. The crystal structure of olmesartan-bound human AT$_1$R (PDB:4ZUD) served as a template to create an inactive apo-AT$_1$R via molecular dynamics simulation (n = 3). All simulations resulted in a water accessible ligand-binding pocket that lacked sodium ions. The model remained inactive displaying little movement in the receptor core; however, helix 8 showed considerable flexibility. A single frame representing the average stable AT$_1$R was used as a template to dock Olmesartan via AutoDock 4.2, MOE, and AutoDock Vina to obtain predicted binding poses and mean Boltzmann weighted average affinity. The docking results did not match the known pose and affinity of Olmesartan. Thus, an optimization protocol was initiated using AutoDock 4.2 that provided more accurate poses and affinity for Olmesartan (n = 6). Atomic models of 103 of the known human AT$_1$R polymorphisms were constructed using the molecular dynamics equilibrated apo-AT$_1$R. Each of the eight ARBs was then docked, using ARB-optimized parameters, to each polymorphic AT$_1$R (n = 6). Although each nsSNP has a negligible effect on the global AT$_1$R structure, most nsSNPs drastically alter a sub-set of ARBs affinity to the AT$_1$R. Alterations within N298 –L314 strongly effected predicted ARB affinity, which aligns with early mutagenesis studies. The current study demonstrates the potential of utilizing in silico approaches towards personalized ARB therapy. The results presented here will guide further biochemical studies and refinement of the model to increase the accuracy of the prediction of ARB resistance in order to increase overall ARB effectiveness.

**Data Availability Statement:** The raw data sets are large, and due to a hard drive crash many of the raw AutoDock output files were lost. However, the PDB used to generate the docking is provided as

supplementary data. The AT1R PDB can be loaded to Model Archive (www.modelarchive.org) after acceptance of the manuscript. The trajectory files are available from Zenodo: The DOI is: 10.5281/zanodo.3988469 And the site is: https://zenodo.org/record/3988469#.Xz1QtehKiHt.

**Funding:** The authors received no specific funding for this work.

**Competing interests:** The authors have declared that no competing interests exist.

## Author summary

The term "personalized medicine" was coined at the turn of the century, but most medicines currently prescribed are based on disease categories and occasionally racial demographics, not personalized attributes. In cardiovascular medicine, the personalization of medication is minimal, despite the fact that not all patients respond equally to common cardiovascular medications. Here we chose one prominent cardiovascular drug target, the angiotensin receptor, and, using computer modeling, created preliminary models of over 100 known alterations to the angiotensin receptor to determine if the alterations changed the ability of clinically used drugs to interact with the angiotensin receptor. The strength of interaction was compared to the wild-type angiotensin receptor, generating a map predicting which alteration affected which drug(s). It is expected that in the future, sequencing of drug targets can be used to compare a patient's result to a map similar to what is provided in this manuscript to choose the optimal medication based on the patient's genetics. Such a process has the potential to facilitate the personalization of current medication therapy.

## Introduction

The Angiotensin (Ang) II type 1 receptor ($AT_1R$) is often studied due to its role in cardiovascular disease, diabetes, and, more recently, cancer [1, 2]. Eight clinically viable antagonists (ARBs) target the $AT_1R$, and ARBs are widely used for the treatment of hypertension, heart failure, and chronic kidney disease. Additionally, due to the prominent role the $AT_1R$ plays in cardiovascular disease, the $AT_1R$ is also the focus of many genetic association studies [3, 4]. However, very few studies have been directed toward pharmacogenomics of the $AT_1R$.

Not all patients respond equally, or at all, to ARBs [5]. One potential reason a patient does not respond, or respond optimally, to an ARB is that there could be a, or many, non-synonymous single nucleotide polymorphism(s) (nsSNP) within the *agtr1* coding sequence. nsSNPs can result in altered antagonist function [6]; thus, as we enter an era dominated by big data and likely personalized medicine, it would be ideal for a prescriber to know which therapies will interact with their target as expected in each patient. Such patient-specific knowledge can come from genetic screening coupled to robust databases linking drug affinities and effects to the genetic sequence of the target receptor(s). Alternatively, if there is no previous data, then there should be a mechanism allowing rapid assessment of which drugs are appropriate for a given patient, such as in silico modeling.

The $AT_1R$ was cloned in the early 1990s and recently was crystallized with antagonist bound [7, 8]. Before crystallization, the ARB binding pocket was investigated primarily through mutagenesis studies [9–12]. These studies identified residues involved in ARB binding that are within the known binding pocket but also identified residues involved in ARB binding that are far from the known binding pocket [9, 10]. Such data demonstrate that single amino acid changes in the $AT_1R$ far from the binding pocket can alter the receptor conformation and disrupt ARB binding. Moreover, in many cases, the mutant $AT_1R$s still bound to, and transduced signals from, Ang II demonstrating the ability of an nsSNP to maintain the physiological functions of the $AT_1R$ yet display ARB resistance. Multiple genomic projects identified polymorphisms within the $AT_1R$ [13], indicating many known polymorphisms may potentially contribute to ARB resistance.

Herein, the Olmesartan bound crystal structure of the $AT_1R$ (PDB: 4ZUD) [7] was used as a template to create a human $AT_1R$ containing residues 1 through 316, which extends through

half of helix 8. The model was then inserted into a membrane consisting of 13% cholesterol [14] and 87% Phosphatidylcholine (POPC) and simulated using all-atom molecular dynamics (MD) in a water box containing 150 mM NaCl for three replicas of 150 nanoseconds (ns). Olmesartan was docked in the binding pocket of an MD-equilibrated structure using three docking programs to compare their docking scores with known experimental values. We then used experimental binding affinities to optimize the docking parameters. Eight clinically used ARBs were docked to the AT$_1$R model. Subsequently, 103 nsSNPs from the 1000 genomes project were individually introduced into the apo-AT$_1$R model, and each system was subjected to energy minimized using Molecular Operating Environment (MOE) software. Each of the eight clinically viable ARBs were docked to each polymorphic AT$_1$R in order to predict which nsSNPs would lead to ARB resistance. We found that most nsSNPs drastically alter a sub-set of ARBs affinity to the AT$_1$R. To our knowledge, this is the first large scale investigation into known human nsSNPs within the AT$_1$R.

## Results

### Validation of the wild-type inactive apo-AT$_1$R model

Crystalized G protein-coupled receptors (GPCRs) often contain a tightly bound ligand that stabilizes a unique ligand-induced conformation. In order to obtain a model of the empty AT$_1$R, the AT$_1$R crystal structure (PDB: 4ZUD) was modified to remove non-receptor residues and the unresolved flexible loops were added back to the receptor, then a short 150 ns MD simulation was conducted within a POPC:cholesterol (87:13 ratio) membrane to relax the structure to a ligand-free state (n = 3); the trajectory files are available at https://zenodo.org/record/3988469#.X2pH4mhKiHu. Each simulation reached a stable structure around 100 ns (S1 Fig.); the last 20 ns of the simulations were considered stable and used for analysis. A single frame representing the average stable root-mean-square deviation (RMSD) from the last 20 ns of replica 1 was chosen as the representative model of the empty AT$_1$R; the PDB file format is available (S1 File). The root-mean-square fluctuation (RMSF) of each protein residue was plotted on the protein backbone and colored based on the degree of movement (Fig 1A). Besides the flexible loops, helix 8 displayed the most fluctuation (RMSF > 3 Å). Expectedly, removal of Olmesartan resulted in movement in the ligand-binding pocket (upper third of the AT$_1$R); however, the remaining core of the receptor displayed little movement (Fig 1A). The RMSD of helix 8 only (S1 Fig.) indicates the position of helix 8 varied by up to 10.6 Å from the empty AT$_1$R model with a mean variation of 8.5 Å ± 2.0 Å over the three replicas, which is similar to previous MD simulations [15]. To further validate the protein conformational ensemble sampled during MD simulations, distance measurements corresponding to double electron-electron resonance (DEER) spectroscopy of the apo-AT$_1$R [15] were measured over the last 20 ns of simulation (Fig 1B). Red Xs in Fig 1B represent the major modes of the DEER data; 50% of the major modes from the DEER experiments are within distances observed in the modeled empty AT$_1$R. Moreover, most of the measurements from the model lie within the range identified in the DEER spectroscopy data. Only two pairs of residues F55-R139 and D236-R311, which Wingler et al. call TM1-TM2 and TM6-helix 8, respectively, do not necessarily represent the DEER spectroscopy [15]. Overall, the data suggest that the MD simulated empty AT$_1$R model is in agreement with the known structural features of the apo-AT$_1$R.

In theory, an active empty receptor conformation could be obtained by removal of the inverse agonist Olmesartan followed by MD simulation, especially since the AT$_1$R is known to display limited basal activity [16]. In order to confirm that the empty AT$_1$R model is in an inactive conformation, consensus GPCR changes [17] between inactive and active structures were measured in matched inactive and active GPCR crystal structures, as well as over the last

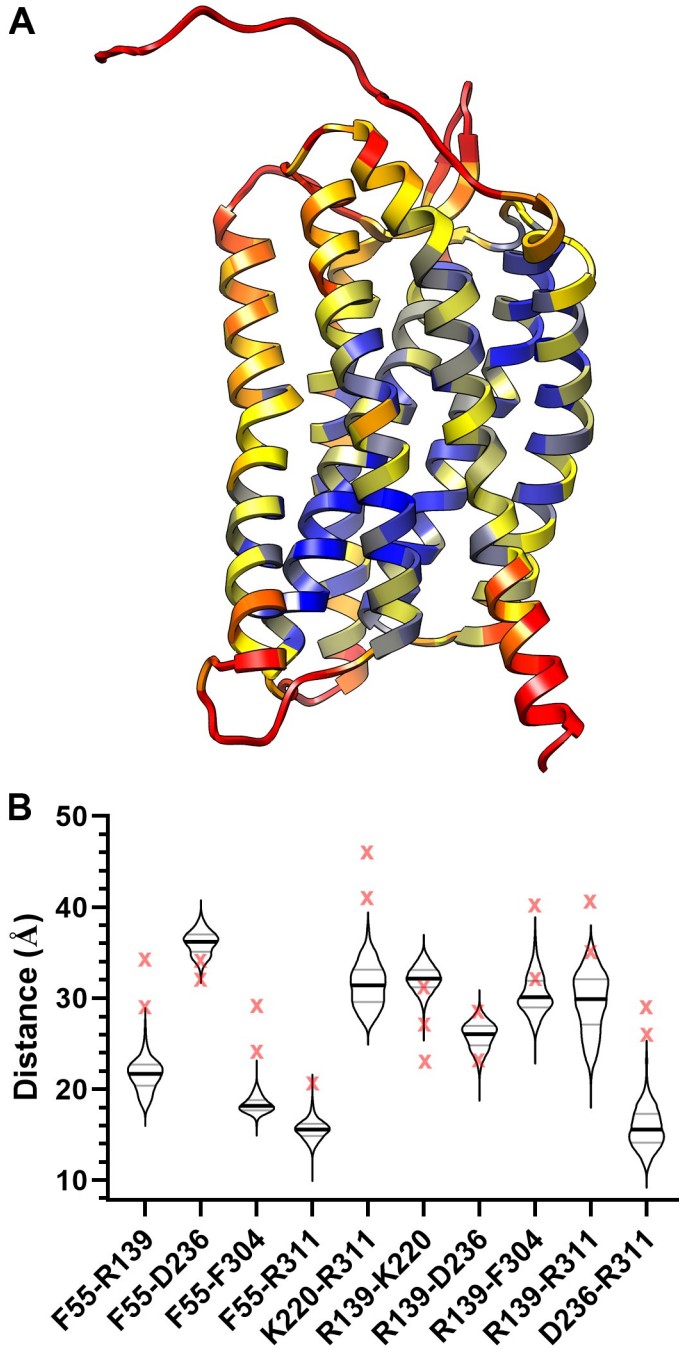

**Fig 1. Apo-AT₁R model. A**, The apo-AT₁R model with the movement of each residue color-coded based on the RMSF. The minimum observed RMSF (0.796 Å) is blue, 25th percentile (1.689 Å) is yellow, median (2.272 Å) is orange, and the 75th percentile (3.165 Å) and higher is red. **B**, A violin plot of the distances (Å) between the center of mass of the residues used in DEER spectroscopy [15] over the last 20 ns of simulation. The thick line represents the median and the thinner lines the 25th and 75th percentile. The red X represents the major modes from the DEER spectroscopy.

20 ns of each simulation (Table 1). No significant conformational change from the crystalized AT₁Rs was observed. However, the distance between residues 1.53 (V49) and 7.53 (Y302) is lower in the AT₁R inactive crystals compared to the reference set by 1.2 Å, which explains the lower value observed in the simulations (Table 1). Similarly, the distance between 5.55 (T213)

**Table 1. Comparison of the apo-AT$_1$R model structure to consensus changes in GPCR structure when activated and the AT$_1$R crystal structures.** All measurements are from the center of mass and are presented in angstroms.

| Interaction | Inactive GPCRs* | | Active GPCRs* | | AT$_1$R model (last 20 ns)[†] | |
|---|---|---|---|---|---|---|
| | Mean | SD | Mean | SD | Mean | SD |
| 3.46 to 6.37 | 6.2 | 0.5 | 11.6 | 1.5 | 6.0 | 0.2 |
| 1.53 to 7.53 | 6.8 | 0.6 | 11.1 | 0.5 | 5.5 | 0.3 |
| 5.55 to 6.41 | 10.6 | 2.0 | 7.4 | 1.5 | 13.3 | 0.4 |
| 3.46 to 7.53 | 11.9 | 0.9 | 7.0 | 0.6 | 12.5 | 0.2 |

* Based on data from A.J Venkatakrishnan et al. [17] the identified residues were measured in inactive (1GZM, 2RH1, 3EML, 3UON, 4DKL, 4YAY, and 4ZUD) and matching active (3PQR, 3SN6, 2YDV, 4MQS, 5C1M, and 6DO1) receptor PDBs; n = 6 and 5, respectively.

[†] The distance was calculated over the last 20 ns for each rep resulting in three average values, which were then averaged; n = 3.

and 6.41 (V246) is greater in the AT$_1$R inactive crystals compared to the reference set by 3.4 Å. Furthermore, structural alignment of the DRY, PIF, and NPxxY motifs between the empty AT$_1$R model and the inactive AT$_1$R bound to ZD7155 (PDB:4YAY) indicate that the activation motifs resulted in no significant changes in orientation of the residues (Fig 2). Similarly, comparison to the inactive A$_{2A}$R crystal structure (PDB:3EML) demonstrates that the empty AT$_1$R model is inactive (S2 Fig.). The NPxxY motif is not as clearly aligned as the DRY and PIF, but this is primarily due to the unique position of helix 8 in the empty AT$_1$R model. For further comparison, the empty AT$_1$R model was aligned to the active AT$_1$R (PDB:6DO1) and mu-opioid receptor (PDB:5C1M). The active AT$_1$R does not display the typical break of the Asp (D)–Arg (R) salt bridge of the DRY motif but demonstrates the change in position of the PIF and NPxxY motifs expected when a GPCR is in the active conformation (S2 Fig.). The active mu-opioid receptor demonstrates the change in position for the DRY, PIF, and NPxxY motifs when a GPCR is active (S2 Fig.). Given that the model of the apo-AT$_1$R structurally resembles the inactive receptors, not active receptors, the data further suggest that the model is an inactive human apo-AT$_1$R.

Since removing Olmesartan resulted in a change in the binding pocket as observed by RMSF (Fig 1A), the integrity and solvent accessibility of the binding pocket of the model AT$_1$R was examined to ensure that it remained a viable model of the inactive human apo-AT$_1$R. The AT$_1$R crystal structures revealed several critical residues for ARB binding, and all clinically viable ARBs were docked to the ZD7155-bound AT$_1$R crystal structure (PDB:4YAY) [8]. As shown in Fig 3A, the residues interacting with all ARBs (shown as sticks) still orientate toward the central pocket. Residues involved in some, but not all, ARB binding (shown as lines) also line the pocket. Therefore, critical residues involved in binding are still oriented in a manner facilitating binding [8]. The size of the binding pocket fluctuated in each replica (Fig 3B). Replica 1, which was used to create the apo-AT$_1$R model (shown as a star in Fig 3B), displayed a stable contraction of the binding pocket from 4ZUD (1505 Å$^3$) to a mean volume of 977 ± 99 Å$^3$ over the last 20 ns of simulation. Replica 2 displayed considerable variation in ligand pocket volume, but during the stable trajectory had a mean volume lower than 4ZUD at 1379 ± 155 Å$^3$. The volume of the third replicate binding pocket varied greatly and likely did not reach equilibrium based on the variation; however, in the last 20 ns of the simulation, the pocket volume expanded to 2344 ± 260 Å$^3$.

Relaxation of the replica 1 binding pocket could collapse the binding pocket making the receptor model invalid. Water and sodium are proposed to enter the binding pocket, and there is growing evidence that sodium plays an allosteric role in many GPCRs [18, 19]. While water molecules were not initially placed within the binding pocket, each replica contained water in

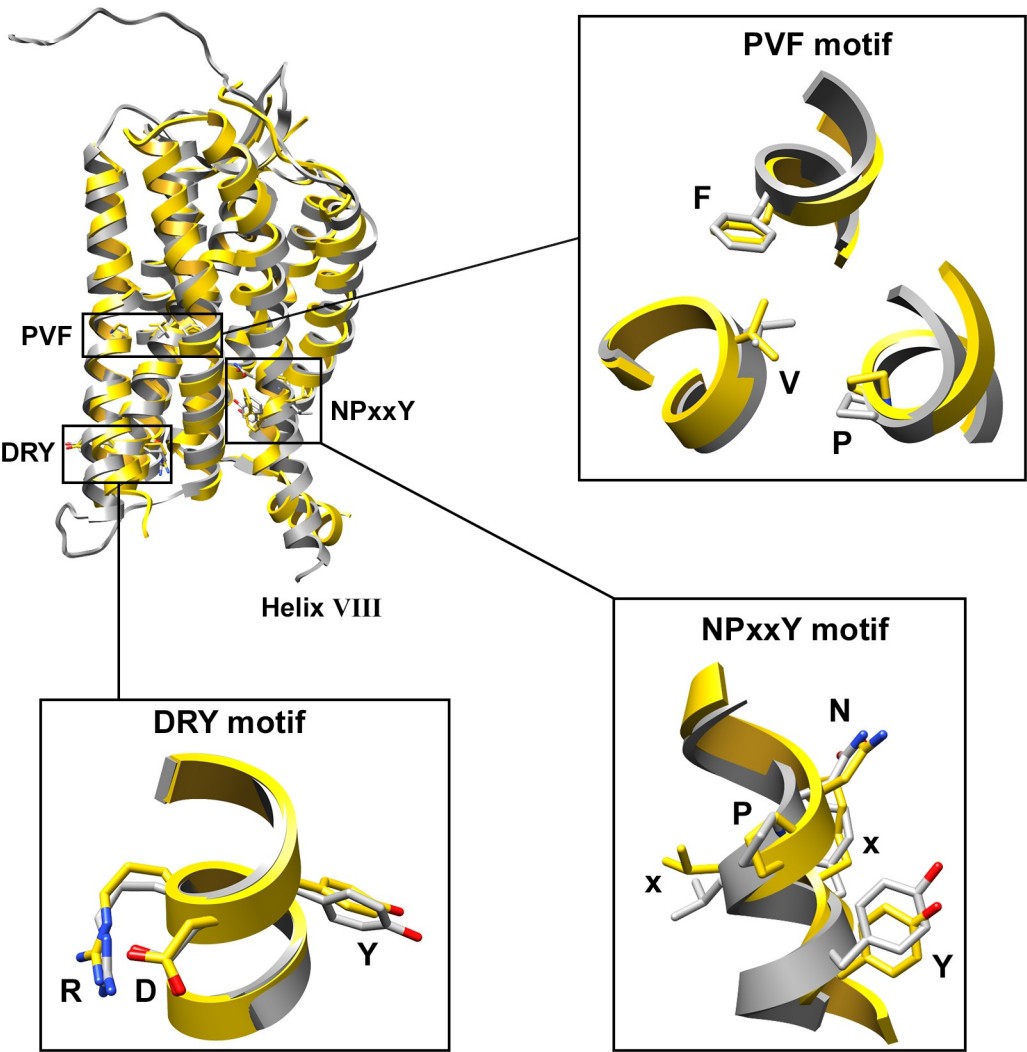

**Fig 2. Comparison of the apo-AT$_1$R model to the inactive AT$_1$R crystal structure (4YAY).** The apo-AT$_1$R model (grey) is aligned to the inactive AT$_1$R (gold), and the activation motifs are expanded in boxes with the residues labeled by single amino acid letter and structure shown as sticks colored by atom type, with the carbons grey and gold for the apo-AT$_1$R model and 4YAY crystal structure, respectively.

the binding pocket (Table 2). The free movement of water in and out of the pocket in each replica indicates no tightly bound water within the pocket, albeit the number of water molecules in the binding pocket remains stable during the simulations. However, sodium was not found in the putative sodium binding pocket nor the ARB binding pocket. Throughout the simulations, the nearest sodium ions were an average of 20.7 Å away from the binding pocket (Table 2). In totality, the data indicates that the model is a valid inactive human apo-AT$_1$R.

## Docking ARBs to the apo-AT$_1$R model

Since Olmesartan was crystalized in the AT$_1$R (PDB: 4ZUD) [7], which served as our starting structure, and was the focus of our previous studies [20], we first docked Olmesartan back into the apo-AT$_1$R model described above (S1 File). Three docking programs were utilized: Auto-Dock 4.2 (Fig 4A), AutoDock Vina (Fig 4B), and MOE (Fig 4C). The predicted poses were

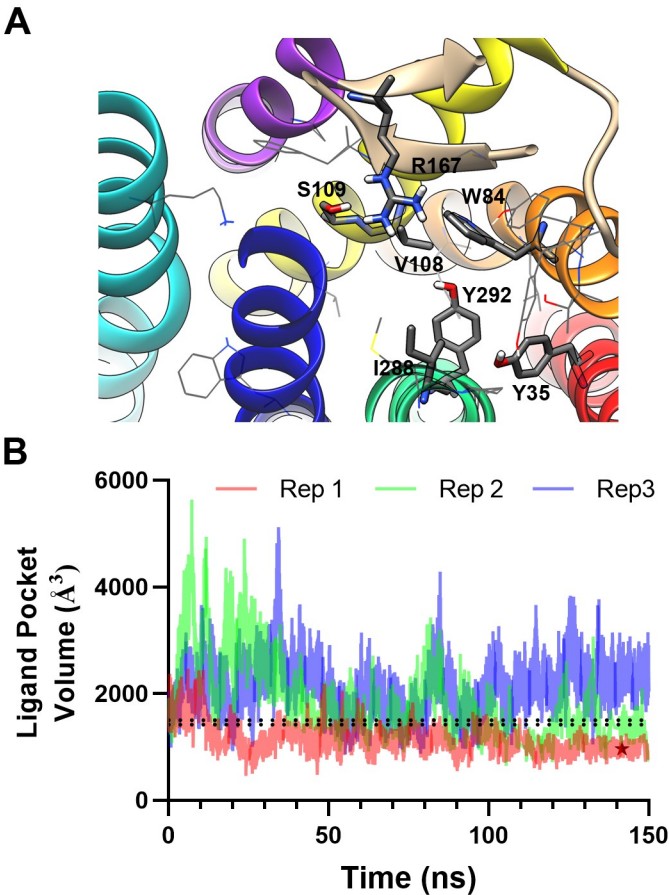

**Fig 3. Apo-AT$_1$R binding pocket. A**, All residues shown to bind to ARBs remain directed toward the ligand-binding pocket in the apo-AT$_1$R model; residues predicted to interact with all ARBs are shown as sticks and residues interacting with a subset of ARBs are shown as lines. However, **B**, the volume of the ligand-binding pocket varied in all three replicas (Rep). The dotted black lines represent the volume of 4ZUD (the upper line at 1505 Å$^3$) and 4YAY (the lower line at 1417 Å$^3$), and the star in replica 1 represents the apo-AT$_1$R model at 961 Å$^3$.

compared to the crystal structure (4ZUD), and the predicted affinity to the experimentally derived 95% confidence interval of the affinity, as expressed in -Log(Molar), of 7.985 to 8.595 [21]. None of the docking programs accurately predicted the affinity, but Autodock 4.2 provided the nearest estimate. AutoDock Vina predicted a single pose somewhat similar to the crystal structure (Fig 4B); however, the predicted affinity was significantly worse than AutoDock 4.2. MOE was run three times, with each iteration containing a change to the system. The first run (grey in Fig 4C) was run identical to the AutoDock runs and produced a single

**Table 2. Solvent in and around the inactive human apo-AT$_1$R.**

| Replica | H$_2$O (TIP3:OH2) within the pocket | | Na$^+$ (SOD) within the pocket | | Na$^+$ (SOD) within 25 Å of Arg 167 | |
|---|---|---|---|---|---|---|
| | # | Total frames (% frames) | # | Total frames (% frames) | # | Mean ± SD distance (Å) |
| 1 | 36 | 4008 (26.7%) | 0 | 0 (0%) | 2 | 21.3 ± 3.3 |
| 2 | 30 | 2513 (16.8%) | 0 | 0 (0%) | 3 | 20.0 ± 3.6 |
| 3 | 32 | 4790 (31.9%) | 0 | 0 (0%) | 3 | 20.8 ± 3.3 |

# indicates the total number of unique atoms in the binding pocket

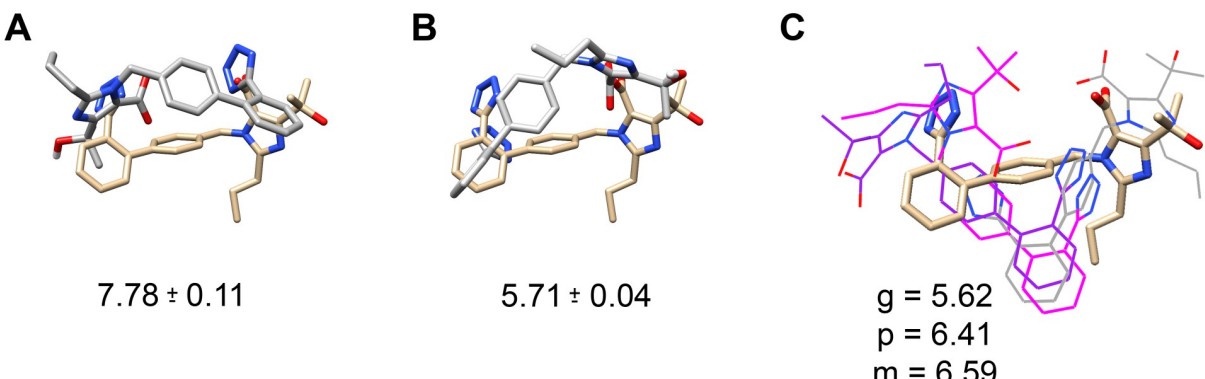

**Fig 4. Docking poses of Olmesartan to the apo-AT$_1$R model. A**, AutoDock 4.2, **B**, AutoDock Vina, and **C**, MOE were used to dock Olmesartan to the apo-AT$_1$R model and compared it to the orientation of Olmesartan in the AT$_1$R crystal structure (4ZUD) shown in tan. The Boltzmann weighted average predicted affinity in -log[M] for each docking run is listed under the image. MOE, **C**, was used to create three different docking scenarios: standard docking (grey), manually setting the tetrazole to -1 charge (purple), and allowing for flexible receptor residues (magenta). For clarity, the MOE dockings are shown as lines.

pose in the correct orientation but not similar to the crystal structure or AutoDock Vina. One problem with AutoDock is that it does not parameterize the tetrazole ring properly as it does not have a net -1 charge; therefore, a net -1 charge was added to the deprotonated tetrazole ring in MOE (purple in Fig 4C) and subsequently a flexible docking program was run (magenta in Fig 4C). The alterations failed to improve the binding pose, in fact the predictions flipped Olmesatan in the binding pocket. Since the affinities of AutoDock 4.2 were closer to the known range and the parameter files could be explicitly controlled, further docking protocol optimization was carried out in AutoDock 4.2.

The inaccurate binding poses, and lower than expected weighted average affinity spurred a pilot project to determine if altering AutoDock 4.2 settings would improve the binding poses and affinities. First, the grid point spacing (GPS) of the search space was systematically altered from 0.375 Å to 0.150 Å, resulting in a non-linear relationship between GPS and Boltzmann weighted binding affinity. Second, the center of the grid box along the z-axis (Z-center) was shifted stepwise by 0.5 Å in both directions resulting in a second, yet unique, non-linear relationship between Z-center and affinity. Therefore, a seven-by-seven matrix obtained at different GPS and Z-axis points was created for Olmesartan. The matrix was then fit to three different mathematical interpolations to identify the GPS and Z-center that provided predicted affinities near the known affinity (see Methods for details and S1 Text for the script). Each matrix derived GPS and Z-center was run through six separate AutoDock 4.2 runs set to provide 100 predicted poses and affinities. The Boltzmann average affinity for each run was arithmetically averaged, and the mean affinity for each interpolation predicted GPS and Z-center were compared to the known Olmesartan values [21]. The cubic interpolation of the Olmesartan seven-by-seven matrix (Fig 5A) provided the most accurate GPS and Z-center (Table 3) and resulted in binding poses that closely matched the crystal structure (Fig 5B). The mean affinity of the Boltzmann weighted averages of each docking run (n = 6) matched known values: the AutoDock 4.2 predicted value for Olmesartan is 8.252 with a 95% confidence interval spanning 8.230 to 8.297, which is within the experimentally derived range of 7.985 to 8.595 [21].

Subsequently, each clinically viable ARB was analyzed via identical GPS and Z-center seven-by-seven matrices and interpolations; only the interpolation providing GPS and Z-centers that resulted in affinities closest to known affinities are shown in S3 Fig. Telmisartan was

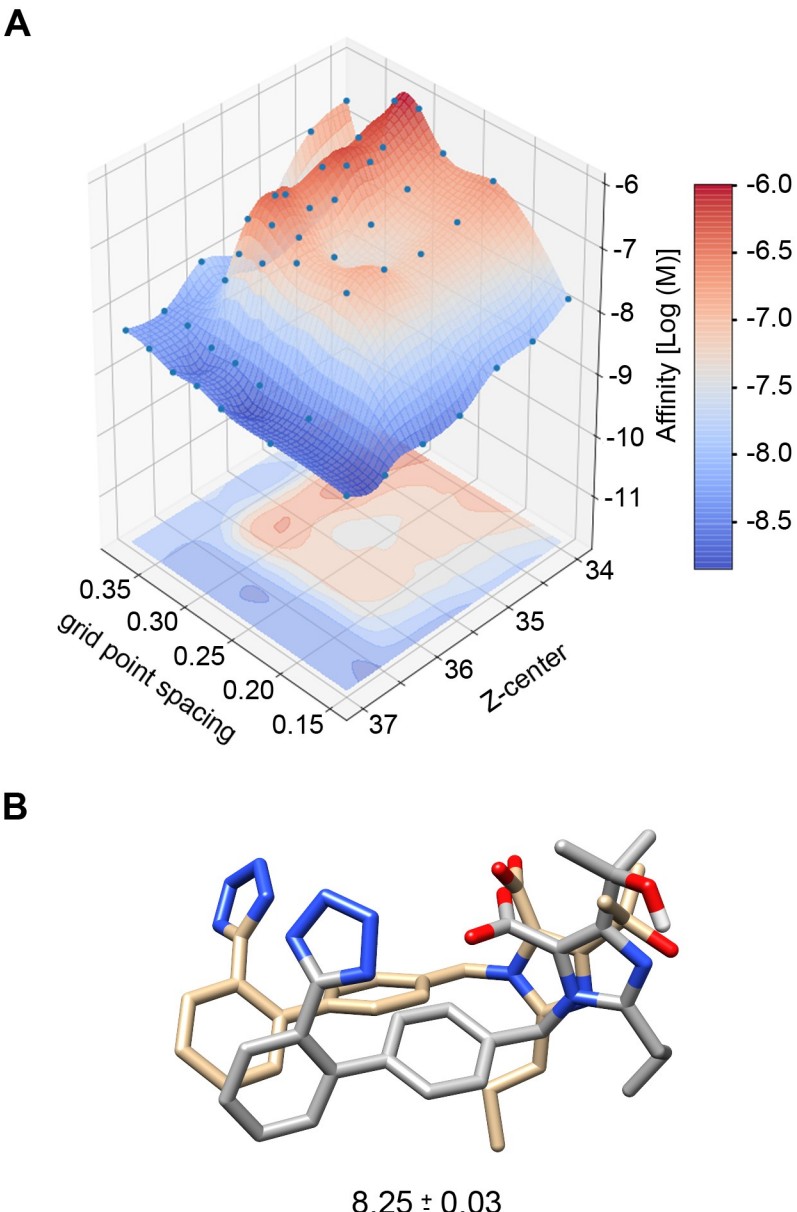

**Fig 5. Optimizing AutoDock 4.2 for Olmesartan binding. A**, 3D fitting to optimize AutoDock 4.2 grid point spacing (GPS) and grid box Z-center for Olmesartan binding to the apo-AT$_1$R model. **B**, The optimized GPS and Z-center produced binding poses (grey) similar to the orientation of Olmesartan in the AT$_1$R crystal structure (4ZUD) shown in tan and generated an accurate Boltzmann weighted average predicted affinity (listed as -log[M]).

unique in that the seven-by-seven matrix identified a GPS and Z-center point that produced the known median affinity; therefore, it is not shown in S3 Fig. The GPS and Z-center parameters optimized for each ARB display remarkable similarity to known affinities (Fig 6); the mean of all the predicted affinities resided within the known affinity range for each ARB (Fig 6 green circles). Whereas, running the same AutoDock 4.2 parameters as were initially used for Olmesartan produced affinities that were outside the range (4 of 8) or outside the middle 50% region (3 of the 4 remaining ARBs) (Fig 6 blue circles). Only Valsartan from the original docking parameters provided values similar to the known median affinity.

**Table 3. Optimal AutoDock 4.2 parameters identified to obtain known median affinity for each ARBs to the apo-AT$_1$R model.**

| ARB | Median Affinity* | | Gridpoint Spacing | X-center | Y-center | Z-center | Number of Points | | |
|---|---|---|---|---|---|---|---|---|---|
| | Log (M) | nM | | | | | X | Y | Z |
| Azilsartan | -8.51 | 3.09 | 0.168 | 38 | 61 | 36.74 | 126 | 126 | 70 |
| Candesartan | -8.46 | 3.47 | 0.342 | 38 | 61 | 34.38 | 50 | 50 | 30 |
| Eprosartan | -8.26 | 5.50 | 0.161 | 38 | 61 | 36.94 | 126 | 126 | 70 |
| Irbesartan | -8.72 | 1.91 | 0.168 | 38 | 61 | 36.84 | 126 | 126 | 70 |
| Losartan (EXP3174) | -7.71 | 19.50 | 0.150 | 38 | 61 | 37.00 | 126 | 126 | 70 |
| Olmesartan | -8.17 | 6.76 | 0.357 | 38 | 61 | 35.55 | 55 | 55 | 30 |
| Telmisartan | -8.17 | 6.76 | 0.275 | 38 | 61 | 36.50 | 70 | 70 | 40 |
| Valsartan | -8.3 | 5.01 | 0.318 | 38 | 61 | 35.33 | 50 | 50 | 30 |

* The median affinity is derived from the data presented in Fig 6, which is from Michel MC et al. [21].

## Docking ARBs to polymorphic AT$_1$Rs

After optimizing the parameters to produce affinities in line with experimentally derived affinities, the *agtr1* nsSNPs from the 1000 genome database were mapped to the AT$_1$R (Fig 7). The polymorphisms were introduced into the inactive apo-AT$_1$R model embedded in a POPC/cholesterol membrane and underwent energy minimization using MOE. Each polymorphic apo-AT$_1$R was aligned to the wild-type empty AT$_1$R, and the ARB specific GPS and Z-center parameters were utilized to conduct docking identical to the wild-type AT$_1$R (Fig 6). The data, summarized in Fig 8, represent predicted affinities reduced by 2-fold or more and that are statistically different from the predicted affinity for the given ARB to the wild-type apo-AT$_1$R. No affinities were statistically increased by 2-fold or more than the control. The data predicts that many polymorphisms alter ARB affinity, but few polymorphisms adversely affect the affinity of all ARBs.

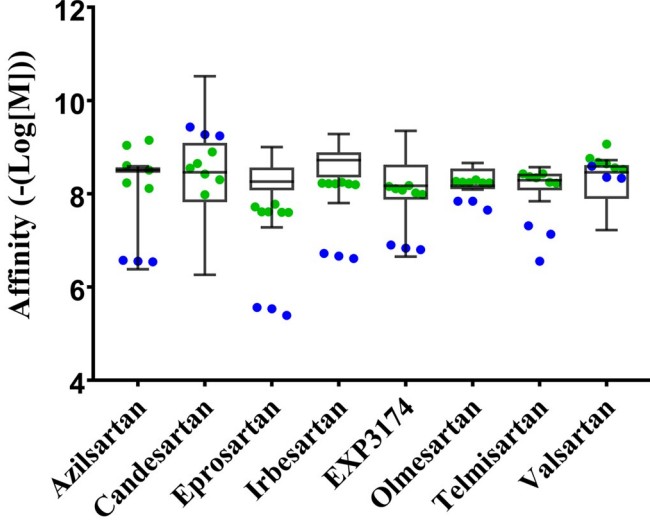

**Fig 6. Docking ARBs to the apo-AT$_1$R model.** Original AutoDock 4.2 parameters (blue points) and optimized docking parameters (green points) are overlaid on a box and whisker plot of experimentally derived ARB affinity [21]. The whiskers represent the maximum and minimum affinity, and the box represents the second and third quartiles with the line representing the median.

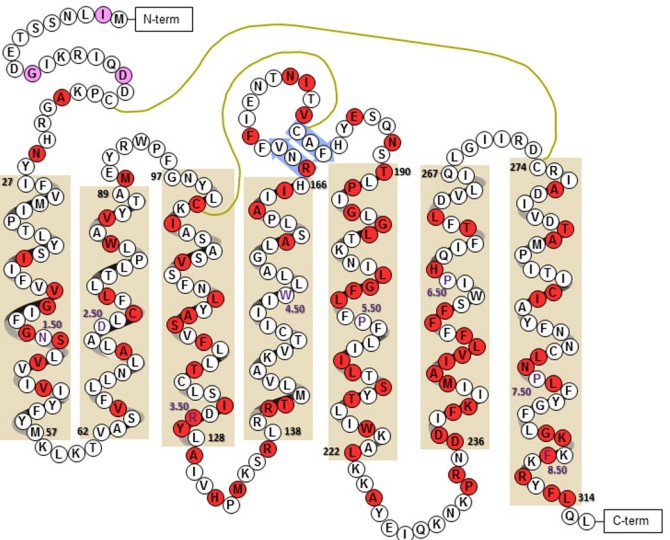

**Fig 7. Location of hAT₁R nsSNPs.** A cartoon of the apo-AT₁R model with alpha helixes depicted within tan boxes, beta sheets depicted within blue arrows, di-sulfide bonds as mustard-colored lines, and residue numbers displaying the residue id of the beginning and end of each helix in black, as well as the Ballesteros-Weinstein numbering of the most conserved residue in purple text. Each polymorphism is shaded pink, if not studied, or red, if examined in this manuscript. The carboxyl unstructured region was not included and contains an additional 15 nsSNPs.

Multiple mutagenesis studies of the AT₁R coupled to radioligand displacement assays have been conducted since the AT₁R was cloned; however, only a few studies utilized mutations that correspond to known nsSNPs of the AT₁R [9, 10], and only one used ARBs that are in clinical use [22]. Arsenault et al. specifically examined the $K_d$ of ARBs to A163T hAT₁R and found particular resistance to Losartan, EXP3174, and Irbesartan, as well as a higher affinity for Telmisartan [22]. The data from Arsenault et al. was transformed into a ratio to compare to the data presented in Fig 8 (Table 4). The predicted affinities correctly predict that EXP3174 and Irbesartan have decreased affinity for A163T AT₁R as well as suggest that Telmisartan binds more readily; however, the predictions for Candesartan and Valsartan are far from the measured values. Three-fifths of the predictions were accurate in terms of the direction of the change in affinity; whereas, a different 3/5ths set of the predictions was accurate within a 2-fold cutoff. Only the predicted values for Irbesartan and Telmisartan were accurate by both measures, and only the prediction of Candesartan affinity was inaccurate by both measures.

## Discussion

### Apo-AT₁R model

The apo-AT₁R model is the third ligand-free molecular dynamics derived model created after the AT₁R was crystallized [15, 23]. In the crystallized inactive AT₁Rs (4ZUD and 4YAY) Arg167 points toward the ligand-binding pocket [7, 8]; however, homology modeling failed to accurately predict the orientation of Arg167 [24, 25]. Arg167 is essential for Ang II binding based on mutagenesis and modeling studies [7, 11, 23, 26] and predicted to be essential for ARB binding based on the crystal structures and subsequent docking [8]. Furthermore, this study also predicts that Arg167 is involved in the binding of most ARBs (Fig 8). Therefore, homology-based AT₁R models that have Arg167 oriented away from the binding pocket are not accurate, and ligand binding to homology models with Arg167 in the wrong orientation are subsequently inaccurate. Based on the orientation of crucial ARB binding residues within

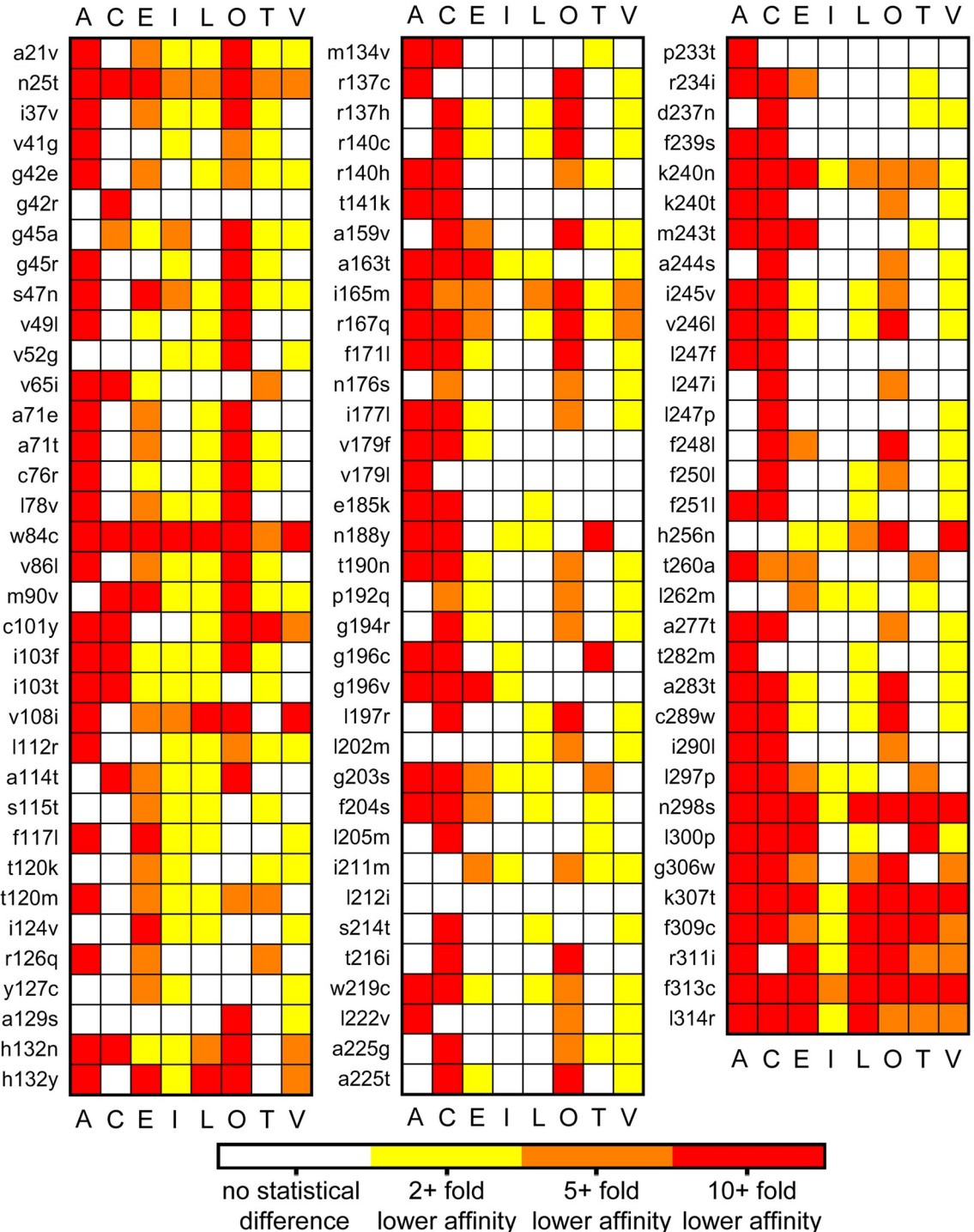

**Fig 8. Heatmap of the predicted effect of hAT$_1$R nsSNPs on ARB affinity.** Each ARB, denoted by the first letter of their name, with EXP3174 marked as L for Losartan, was docked to single polymorphisms induced into the apo-AT$_1$R model. The polymorphism is listed on the left of each column. Data are colored, as per the key, only if there is a statistical difference between the Boltzmann weighted average affinities (n = 6) of the apo-AT$_1$R model and indicated polymorphism as determined by a one-way ANOVA followed by Kruskal-Wallis Multiple-Comparison Z-Value Test.

**Table 4. Comparison of experimental and docking affinities to polymorphic AT$_1$R.**

| nsSNP (Ref) | ARB used experimentally | Experimental Ratio* | Optimized Parameters Docking Ratio* | Within 2-fold the error? |
|---|---|---|---|---|
| A163T [22] | Candesartan | 0.94 ± 0.09 | 21.24 ± 16.52 | No |
| | Irbesartan | 1.84 ± 0.22 | 2.52 ± 0.14 | Yes |
| | EXP3174 | 7.81 ± 1.12 | 3.43 ± 0.69 | No |
| | Telmisartan | 0.70 ± 0.14 | 0.50 ± 0.15 | Yes |
| | Valsartan | 1.10 ± 0.22 | 3.25 ± 1.21 | Yes |

* Ratios were determined by dividing the reported data for nsSNP by wild-type, and error was calculated through error propagation using the standard deviation.

the apo-AT$_1$R (Fig 3A), the ligand-binding pocket of the empty AT$_1$R model is likely an accurate model of the ligand-free AT$_1$R.

Although the apo-AT$_1$R model is in the inactive state (Table 1 and Fig 2), the orientation of helix 8 in the apo-AT$_1$R model was surprising due to its amalgamation with helix 7 (Fig 1A), which proved to be a single frame in a highly flexible helix 8 (S1 Fig.). Helix 8 is not present in 4ZUD [7], but is in 4YAY [8] as well as the active AT$_1$R structure 6DO1 [26]. The AT$_1$R crystal structures place helix 8 slightly bent from the membrane and parallel to the membrane, respectively. Fortunately, the mobility of helix 8 is similar to what was measured via DEER spectroscopy and MD modeling of the apo-AT$_1$R, suggesting that it is not purely an artifact [15, 23]. Helix 8 is flexible without negatively charged phospholipids [15]; however, the flexibility of helix 8 is likely restrained in vivo due to the C-terminus anchoring to negative phospholipids in the membrane [23, 27–29]. The systems presented here lacks lipids shown to enhance helix 8 association with the membrane as the modeled membrane only contains POPC and cholesterol [27–29]. Additionally, 4YAY, which served as a guide, does not contain the entirety of helix 8; thus, the apo-AT$_1$R model lacks four positively charged amino acids that facilitate binding to negative phospholipids [29]. Therefore, the mobility of helix 8 in the apo-AT$_1$R model may be dependent on its length and the lack of negative phospholipids. Future MD simulations of the AT$_1$R, and likely other Gq-coupled GPCRs, should be conducted in a membrane with PIP2 as PIP2 is the substrate of PLC and is expected near Gq-coupled GPCRs.

Ligand binding pockets of GPCRs must be solvent assessable, for that is the only way that ligands and sodium can enter the pocket. Sodium is described as an allosteric modulator of GPCRs [18, 19]; however, the role of sodium as an allosteric modulator of the AT$_1$R is debatable because low sodium that is an order of magnitude below physiological levels has little effect on the wild-type AT$_1$R but strongly induces activity in mutant AT$_1$Rs [30]. Regardless of the role sodium plays in the AT$_1$R, the residues creating a sodium pocket are present in the AT$_1$R [7]. The potential role of sodium in the function of the AT$_1$R and the requirement of an appropriate model to allow for water movement in and out of the binding pocket spurred the investigation into water and sodium within the apo-AT$_1$R ligand-binding pocket (Table 2). During the simulations, water moves into and out of the ligand-binding pocket, and the ligand-binding pocket contains water once the simulation reached equilibrium. Strikingly sodium does not approach the apo-AT$_1$R. The positive residues in the binding pocket may exclude sodium from entering the pocket with water; however, this explanation alone does not explain the distance the sodium maintained for the duration of all the simulations. Future studies are required to determine if sodium can enter the AT$_1$R or if sodium ions are encapsulated when a GPCR folds into its canonical shape.

## Docking ARBs to the apo-AT$_1$R

Given that the ligand-binding site of the empty AT$_1$R model appears appropriate, all ARBs were initially docked using AutoDock 4.2 (Fig 6 blue circles). The results were not ideal as the Olmesartan binding poses were not accurate; (Fig 4A) the predicted affinity for half the ARBs was outside experimentally measured values, and 75% of the ARBs predicted affinity were far from the median experimental value. AutoDock Vina and MOE were also used to investigate ARB binding using Olmesartan as a model ARB (Fig 4). MOE failed to predict an appropriate pose and affinity, and Vina obtained a likely accurate pose but could not predict the affinity. The problem with using AutoDock 4.2 and AutoDock Vina is that the programs rely on the same ligand parameter file, and AutoDock parameterization of the tetrazole ring in Olmesartan is not a net negative one charge as it should be. Therefore, it is not surprising that the affinity and many poses are not accurate when Arg167 is a crucial residue for binding as it coordinates the two net negative charges on Olmesartan [7]. Additionally, the apo-AT$_1$R model ligand-binding pocket is only 961 Å$^3$, which is 63% the size of 4ZUD (1505 Å$^3$); (Fig 3B) consequently excluding the possibility that Olmesartan, and by extension the other ARBs, will bind in the same orientation as the crystal structures. Additionally, given that the magnitude of the apo-AT$_1$R model ligand-binding pocket volume reduction, using flexible residues would not suffice to mimic induced fit as exemplified by the MOE trial (Fig 4C).

Since AutoDock 4.2 provided the highest affinity (Fig 4A) and had more discrete control over the docking parameters, AutoDock 4.2 was chosen to continue docking. However, the Boltzmann weighted average affinity for each docking run for all of the ARBs, except for Valsartan, were not optimal (Fig 6). Therefore, we began to explore the input files to determine what could be changed to improve Olmesartan binding. The first parameter we changed was the Z-center. The Z-center was moved by 0.5 Å keeping the search space within the ligand-binding pocket as defined by the crystal structure [7]. Altering the Z-center altered the Boltzmann weighted average predicted affinity non-linearly. Secondly, the GPS was altered while keeping the search space within the binding pocket, and similarly, the Boltzmann weighted average predicted affinity changed in a non-linear manner. Therefore, curve fitting (Fig 5A and S3 Fig.) was used to predict the optimal Z-center and GPS for each ARB (Table 3). Using Olmesartan as a model ARB, the optimized GPS and Z-center resulted in the most accurate binding pose (Fig 5B) and affinity that matched the experimentally derived median affinity (Fig 6). The predicted affinity for all ARBs, except for Valsartan, were closer to their respective experimentally derived affinity (Fig 6).

Changing the Z-center and GPS only changed the search space, it did not fix the aforementioned parameterization problem. Such an approach may be able to be used for other receptors but should only be used if there is an experimentally derived affinity. The study by Michel et al. [21] is a meta-analysis of many studies providing a solid benchmark to optimize docking. The ARB-specific parameter optimization method presented here can be used to obtain affinities that are greater than the known range; thus, such protocols should be used with care. Changing the Z-center and GPS did improve Olmesartan poses within the AT$_1$R and more poses matched the crystal structure (Fig 5B), but most of the 100 poses did not match the crystal structure. For this project, the affinity, not binding pose, was the primary objective. With a newly shaped binding pocket (Fig 3B), that is 37% smaller than the crystal structure, it is not reasonable to use previous docking results to a crystal structure formed with the relatively bulky ZD7155 [8] to gage the poses of the ARBs as the reduction of the binding pocket is bound to induce some restrictions. Secondly, the crystal structures provide the lowest energy binding pose in the conditions of the crystal structure; this may not be the only pose an ARB can assume in the AT$_1$R at physiological conditions. Therefore, the Boltzmann weighted

average of each docking run was utilized because it favors the lowest energy (highest affinity) binding pose, which mimics our understanding of pharmacology.

## Docking ARBs to polymorphic AT$_1$Rs

With predicted ARB affinities that were within the range of experimentally derived ARB affinities, large scale docking began to assess many of the known *agtr1* nsSNPs (Fig 7). Unlike traditional docking experiments that target one receptor using a few frames from a simulation with multiple ligands, this study reverses the standard paradigm by examining a few ligands to multiple variations of the same receptor. Consequently, only a single frame from the simulation was utilized to create 103 polymorphic AT$_1$Rs and subsequently use the ARB optimized docking parameters to predict ARB affinity to each known polymorphism. Reliance on only one snapshot of a structure to predict binding has limitations because the possible interactions are greatly constrained by a static space that is generally varied by using multiple frames from a simulation. However, this study is unique as it represents the first examination of nearly all the identified single human nsSNPs within the AT$_1$R.

Although, only Arsenault et al. specifically examined a known polymorphism with ARBs used in this study and the data correlate 3/5$^{ths}$ of the time [22]; the data presented in Fig 8 can be indirectly compared to the residues involved in docking [8] and regions that displayed reduced Losartan binding [9, 10]. Residues predicted to be involved in ARB binding from docking to the crystal structure that are sites of polymorphisms include W84, V108, L122, A163, R167, and H256. The polymorphism W84C is predicted to ablate ARB binding; whereas, the other listed polymorphisms have at least one ARB that is predicted to retain a relatively normal affinity for the receptor (Fig 8). The original mutagenesis experiments designed to identify residues involved in Losartan binding to the AT$_1$R identified residues in the binding pocket as well as additional residues including P192, F248, and L300 [9, 10]. Mutation of P192 and F248 resulted in a 2-fold change in Losartan affinity [10], and our data with different "mutations" corresponds showing minor changes, except for candesartan with F248L, in only half of the docked ARBs (Fig 8). L300, as well as residue F301, which are far from the ligand-binding pocket, displayed a 5-fold decrease in Losartan affinity but no change in the affinity of an Ang II analog when mutated to a more bulky and less bulky residue, respectively [9]; similarly all polymorphisms modeled from N298S to L314R showed drastic inhibition of ARB binding (Fig 8). Within N298 to L314, there are only five out of 64 instances of an ARB displaying affinity that is not statistically different from the control. No other eight consecutive polymorphisms show such a degree of change in predicted affinity. Overall, the docking to polymorphic AT$_1$Rs mirrors previous docking and ligand binding data suggesting that known polymorphisms can alter the affinity for specific ARBs while having a negligible effect on other ARBs (Fig 8).

## Role of molecular modeling in personalized medicine

Achievable personalized medicine can occur through matching the pharmacopeia to a person's genome. For example, polymorphisms within metabolic enzymes are accepted measures to adjust the dosing of a drug or inform clinicians to use an alternative medication to avoid a potential drug-gene interaction. The next steps to bring personalized medicine to the clinic include understanding how polymorphisms alter individual drug effects. Polymorphisms can alter the affinity of a drug as well as alter signal transduction, as exemplified by the example of V302I μ-opioid receptors that results in the ablation of naloxone function [6]. Herein, the data suggest that polymorphisms in the AT$_1$R may be responsible for apparent ARB resistance. Moreover, the data suggests that often there is an ideal ARB for a particular polymorphism; thus, matching a patient's genotype to an existing drug fulfills the promise of personalized medicine.

Using computational modeling provides a relatively inexpensive and rapid drug screening process. As technology progresses, the speed of the computations increases; whereas, fundamental pharmacological and biochemical determination of ligand affinity has changed little, and radio-labeled ligands are not always available. Importantly, the known nsSNPs of the $AT_1R$ examined are only the tip of the proverbial iceberg. There is little data regarding the combination of nsSNPs, and with 103 polymorphisms, the combinatorial potential is staggering. Moreover, Hauser et al. reported that a fraction over 1 in 293 births contain a new nsSNP in a clinically targeted GPCR [6]. Currently, there are an estimated 250 births worldwide per minute; thus, nearly every minute, there is a new polymorphism created in one of the 108 GPCRs that are clinical targets. Therefore, the number of seemingly benign nsSNPs in the $AT_1R$ that may attenuate ARB affinity will likely increase in the future. A computational approach to ARB affinity is a reasonable and comparatively rapid method to tailor ARB therapy to a patient. Ideally, a patient's *agrt1* would be sequenced and compared to a database to select the best ARB. Any novel nsSNPs, or nsSNP combinations, would then be modeled to predict which ARB is most appropriate and, after verification, added to the database.

## Limitations of the current study

Although the affinity estimates for each ARB blinding to the empty $AT_1R$ model are within known ranges and comparison of relative affinities obtained through molecular modeling is standard practice, there are caveats in the experimental design that prevents taking the data directly to the clinic. The caveats include a lack of $AT_1R$ flexibility in the docking algorithm; lack of data on Ang II affinity to compare to ARB affinity, as ARB efficacy is dependent on the ratio of the ARB to Ang II affinity; and lack of a biological understanding of each polymorphism. AutoDock 4.2 allows for limited flexible residues, but do not account for the flexibility of the entire receptor [31]. Since the apo-$AT_1R$ model ligand-binding pocket is 544 Å$^3$ smaller than the ARB bound crystal structures (Fig 3B), only MD simulation will capture the ligand-bound structure and thus more accurately predict affinity. Ang II has too many rotatable bonds to dock reliably in AutoDock 4.2 but can be docked in AutoDock Vina; however, the small pocket of the apo-$AT_1R$ and lack of induced fit in the docking programs will not allow for an accurate prediction of Ang II binding pose or affinity to each polymorphism. As stated, MD simulation allows for all-atom flexibility and estimation of ARB and Ang II affinity, but the computational time required to examine each ligand and polymorphism is beyond the scope of a single laboratory. Additionally, polymorphisms can have a myriad of effects such as altering the surface expression or blocking coupling to the G-protein. Additionally, GPCRs can function as monomers, homodimers, and heterodimers, and the effect polymorphism have regarding dimerization is unknown. Therefore, biological experiments are necessary to establish the functional relevance of each polymorphism as some polymorphisms may ablate the function of the $AT_1R$, thus negating the need for an ARB regardless of predicted affinity. Consequently, to realize personalized drug therapy, modeling and basic laboratories should collaborate to create viable databases to guide clinical decisions. Concurrently, patients should also be tacked to determine if the database is accurate as well as provide a scan for novel polymorphisms or combinations of polymorphisms.

## Methods

### $AT_1R$ model preparation

The crystal structure of human $AT_1R$ bound to olmesartan (PDB: 4ZUD) was downloaded from the RCSB Protein Data Bank [7]. 4ZUD contains apocytochrome $b_{562}RIL$ fused to the amino terminus, and many of the flexible regions, as well as helix 8, are not resolved. In order

to generate an appropriate starting structure, olmesartan and the apocytochrome $b_{562}$RIL fusion were removed from 4ZUD, and the missing regions were added to the protein with MOE software (Chemical Computing Group ULC, Montreal, Canada). Specifically, the N-Terminus (residues 1 to 25), intracellular loop 2 (residues 134 to 140), extracellular loop 2 (residues 186 to 188), intracellular loop 3 (residues 223 to 234), and helix 8 (residues 305 to 316) were added to the $AT_1R$ in accordance to the human $AT_1R$ sequence and PDB:4YAY. The remaining carboxyl-tail of the $AT_1R$ (residues 317 to 359) was not modeled. The $AT_1R$ model then underwent an energy minimization within MOE using the Amber10:Extended Huckel Theory (EHT) force field [32].

## Molecular dynamic (MD) simulations and analysis

The MOE minimized $AT_1R$ was loaded into CHARMM-GUI [33]. An 80 Å by 80 Å lipid bi-layer composed of 13% cholesterol [14] and 87% Phosphatidylcholine (POPC) was generated around the receptor [33]. Water was packed 17.5 Å above and below the lipid bi-layer, and 150 mM $Na^+$ and $Cl^-$ ions were added to the system via Monte-Carlo ion placing. The all-atom CHARMM C36 force field [34] for proteins and ions, and the CHARMM TIP3P force field [35] for water were selected. A hard non-bonded cutoff of 8.0 angstroms was utilized. All molecular dynamics simulations were performed using the PMEMD module of the AMBER16 package [36] with support for MPI multi-process control and GPU acceleration code. Orthorhombic periodic boundary conditions with a constant pressure of 1 atm was set via the NPT ensemble and temperature was set to 310.15˚K (37˚C) using Langevin dynamics. The SHAKE algorithm was used to constrain bonds containing hydrogens. The dynamics were propagated using Langevin dynamics with Langevin damping coefficient of 1 $ps^{-1}$ and a time step of 2 fs. Before the production run, the $AT_1R$ model was minimized for 5000 steps using the steepest descent method and then equilibrated for 600 ps. The protein coordinates were saved in 10 ps intervals. The production run lasted 150 ns, at which point all three replicas were stable for at least 20 ns. The frame representing the value closest to the average RMSD of the stable 20 ns from replica 1 was selected as the structure representing the empty conformation of the $AT_1R$ and herein is called the apo-$AT_1R$ model.

Since the $AT_1R$ has been extensively studied and is known to display basal activity in the absence of agonist [37], the empty $AT_1R$ model was examined to determine if it resembled an inactive or active GPCR. Venkatakrishnam et al. identified residues that are commonly in contact when a GPCR is inactive but not active and vice versa [17]; based on the identified interactions within the crystal structures, UCSF Chimera [38] was utilized to calculate the center of mass and distance between each identified pair (Table 1). CPPTRAJ [39] was utilized to measure the distances between the center of mass of the paired residues in the last 20 ns of simulation time. Wingler et al. conducted DEER spectroscopy of apo-$AT_1R$ [15], similar measurements were made via measuring the distance between the center of mass of the residues examined in the DEER spectroscopy studies across the last 20 ns of simulation time utilizing CPPTRAJ.

Movement of the receptor and water during the simulation were also calculated using CPPTRAJ. The RMSF of the receptor was calculated over the entirety of each trajectory. The RMSF values from each trajectory were averaged then saved as the B-factor and visualized in UCSF Chimera. The RMSD of each frame compared to the starting structure was calculated for the entire trajectory using only the backbone atoms of the seven-transmembrane helixes (helix 1: N25-M57, helix 2: V62-A89, helix 3: N98-I103, helix 4: L138-H166, helix 5: T190-K223, helix 6: D236-L268, helix 7: C274-L305) due to the inherent flexibility of the loops. The RMSD of helix 8 was independently calculated over the entire trajectory using the apo-$AT_1R$ PDB file as a reference and residues G306-L316.

Solvent accessibility to the binding pocket was determined by first identifying residues that line the pocket (S4 Fig.) and the specific atoms of the residues that line the pocket. The residue and atoms utilized were T88:HG1, Y92:HE1, S109:OG, R167:NH2, K199:HZ2, H256:HE1, V264:CG2, M284:CE, and Y292:HH and distances in the PDB file were calculated in PyMol to create overlapping spheres to measure water and sodium. Arg 167 was given an 11 Å radius, and all others a radius, in angstroms, that rarely reach outside of the binding pocket (T88: HG1@4, Y92:HE1@5, S109:OG@5, K199:HZ2@5, H256:HE1@5, V264:CG2@5, M284:CE@7, and Y292:HH@7). Nativecontacts in CPPTRAJ was used to identify the OH2 of TIP3 and sodium molecules within the radii provided, and only OH2 and sodium found within Arg 167 and a second radii were considered in the binding pocket. Sodium and water were also measured in the initial PDB produced by CHARMM-GUI to determine if water entered the ligand-binding pocket. Additionally, sodium was measured from within 5 Å of N295, which is part of the punitive sodium binding site [7]. Because Sodium was not found in the ligand or sodium pocket, the radius of Arg 167 was increased to 25 Å to determine if any sodium ions approached the extracellular side of the receptor.

The volume of the ligand pocket was calculated with the script provided (S2 Text). Briefly, the residues mentioned above lining the edge of the apo-AT$_1$R binding pocket were used to calculate the center of the binding pocket, and a minimum radius of a sphere was calculated to fill the pocket (centroid sphere). Subsequently, 8 Å radii were applied to each of the residue atoms, mentioned above, that line the binding pocket (residue sphere). A breadth-first search algorithm was applied to begin at the center of the centroid sphere and then map the intersection of the centroid sphere and at least one on the residue spheres in cubic angstroms (S4 Fig.). This process was iterated over each frame of each replica, and the data plotted as cubic angstroms volume over time. The two inactive AT$_1$R crystal structures (4ZUD and 4YAY) were measured as controls after creating systems in CHARMM-GUI as described to generate a PDB file within a lipid membrane and the parm7 files. Importantly, the lipid membrane laterally constrains the receptor preventing the breadth-width search from erroneously identifying space outside of the receptor.

## Docking and analysis of docking

Each ARB was initially generated in ArgusLab, then refined in MOE. The metabolite of Losartan, EXP3174, has greater potency and plasma half-life than Losartan [40]; therefore, EXP3174 was utilized instead of Losartan. PDB files and Mol2 files were created in MOE. The PDB of Olmesartan was used to dock to the apo-AT$_1$R in MOE after first aligning Olmesartan and the apo-AT$_1$R model to the crystal structure (PDB: 4ZUD). The Mol2 files were loaded individually into AutoDock Tools 1.5.6. The loaded ARB was then identified as the ligand, then Gasteiger charges were computed, and all atoms were assigned an AD4 type. Ligand torsions were calculated, and the ARB was saved as the PDBQT file type, which is used by AutoDock 4.2 and AutoDock Vina. Importantly, all ARBs with tetrazole rings were parametrized with a deprotonated tetrazole ring.

Independent from the ligands, the apo-AT$_1$R model was loaded into AutoDock Tools 1.5.6, and hydrogens were added to the structure where appropriate. Next, the Gasteiger charges were computed, and then all non-polar hydrogens on the protein were merged before assigning AD4 atom types. The receptor was then saved as the PDBQT file type. Next, the GPF and DPF files were created. The GPF contains the grid point spacing and grid box size and is created via loading the receptor PDBQT file and identifying it as the macromolecule under the grid tab, and then the ligand was loaded and selected under the grid tab. Finally, the grid box size and location were chosen. The initial size and location were based on the location of

Olmesartan in PDB:4ZUD and generated a box that was 57 x 60 x 45 (xyz) with grid point spacing (GPS) at the default 0.2. The GPF file was then saved with the selected parameters. The DPF file was generated by loading the receptor PDBQT file, choosing the ligand, and setting the output to Lamarckian GA then saving the DPF file. The DPF file was edited in a text editor to set the number of runs from 10 to 100, and AutoDock 4.2 was utilized to generate docking poses and predicted affinity.

For AutoDock Vina, the Olmesartan and apo-$AT_1R$ PDBQT files were loaded, and the center and size of the box were converted to angstroms; the energy range was set to 3, exhaustiveness to 8, and the number of modes to 100; however, Vina only provided a maximum of 13 binding poses. For MOE, an iterative binding approach was utilized the Amber10:EHT force field. First, the apo-$AT_1R$ and Olmesartan were separately aligned to 4ZUD, and docking was run for 10 interactions; however, MOE added the hydrogen back to the tetrazole ring and only provided five binding poses. Second, the deprotonated tetrazole ring was forcibly given a negative one charge on the deprotonated nitrogen, and docking was repeated. Third, the flexible residues option was chosen before docking.

The affinities provided by the docking software were in kcal/mol, and each pose has a different affinity. To not bias affinity measurements, a Boltzmann weighted average affinity was calculated from each pose in a single run. The binding free energy ($\Delta G$) in kcal/mol was converted to a predicted dissociation constant ($K_d$) via Eq 1, shown below. Note ($i$) is the index for any given affinity from a docking program, and R is the molar gas constant, and T is the temperature in kelvin at body temperature (310.15).

$$K_d = \frac{1}{e^{\left(-1 \times \frac{\sum_{i=1}^{100} \Delta G_i}{RT}\right)}} \tag{1}$$

Because preliminary studies did not produce affinities near known values, we altered the Z-axis and GPS and found a non-linear relationship regarding the predicted affinity of Olmesartan. Therefore, each ARB was docked to the apo-$AT_1R$ after creating new GPF files with the center position of the grid box on the Z-axis, altered by 0.5 Å in the positive and negative direction by three steps from 35.5 (34.0, 34.5, 35.0, 35.5, 36.0, 36.5, and 37.0). Subsequently each Z-center was given a new GPS (0.150, 0.225, 0.250, 0.275, 0.300, 0.325, 0.350, and 0.375). For each ARB, there were now 49 different GPF files that were run, creating ARB specific 7x7 matrixes of predicted affinity based on different grid locations and spacing. The data for each ARB was converted to pLog affinity (nM) then fit by a nearest, linear, cubic, and bivariate spline 3D-fitting using python (see S1 Text for the python script). The variables resulting in the greatest predicted $K_d$, as well as the known median ARB affinity [21], were extracted from the curve fitting and tested in Autodock 4.2 three times to confirm the predicted values. For Telmisartan, the seven-by-seven matrix identified ideal parameters; thus, those parameters were utilized. The grid box spacing and Z-axis that produced a predicted $K_d$ closest to the experimentally derived median affinity for each ARB was run six times in Autodock 4.2; only data from the most accurate method is reported. See Table 4 for the parameters that produced a predicted $K_d$ similar to the known median affinity for each ARB.

## Generating nsSNPs $AT_1R$ and docking ARBs

Data from the 1000s genome project was mined to extract all non-synonymous single point mutations (nsSNPs) [13]. The empty $AT_1R$ within the lipid bilayer was loaded into MOE, and each of the 103 chosen nsSNPs was generated individually using the protein builder function in MOE software. After changing the residue, an energy minimization utilizing Amber10 EHT

force field was conducted via MOE to generate a conformation of the $AT_1R$ carrying the polymorphism. The atom coordinates of empty polymorphic $AT_1R$ were saved as a PDB and aligned to the wild-type empty $AT_1R$ so that all docking coordinates would be similar; thus, allowing for the individual ARB optimized parameters to be used on each polymorphism. Each polymorphism was loaded into AutoDock Tools 1.5.6 to prepare the receptor PDBQT file, as explained above. Each ligand was docked to each receptor 100 times in a single Auto-Dock 4.2 run, and each run was replicated six times. The affinities were extracted using a script and processed to determine the affinity as described above.

## Statistical analysis

The fold change in binding affinity between wild-type $AT_1R$ and each polymorphic $AT_1R$ was analyzed within NCSS 2007 statistical software (Kaysville, UT) using a one-way ANOVA followed by Kruskal-Wallis Multiple-Comparison Z-Value Test. Data are shown if the fold change is 2-fold or higher and statistically different than control. No ARBs displayed a reduction by 2-fold or higher and were statistically different from control.

## Supporting information

**S1 Fig. Analysis of structural stability and helix 8 movements. A**, RMSD of each model indicates that the empty $AT_1R$ models are stable; **B**, however, helix 8 is mobile when global movement is minimal. The apo-$AT_1R$ model served as the reference for helix 8 mobility, and it is the only frame that orientates helix 8 as an extension of helix 7 (0 RMSD in replica 1 red line R1:h8).
(TIF)

**S2 Fig. Comparison of the apo-$AT_1R$ model to inactive $A_{2A}R$ (3EML), active $AT_1R$ (6DO1), and active μOR (5C1M) crystal structures.** The apo-$AT_1R$ model (grey) is aligned to the, **A**, inactive $A_{2A}R$, **B**, active $AT_1R$, and, **C**, active μOR crystal structures (gold). The activation motifs are expanded in boxes with the residues labeled by single amino acid letter and structure shown as sticks colored by atom type, with the carbons grey and gold for the apo-$AT_1R$ model and comparative crystal structure, respectively.
(PDF)

**S3 Fig. ARB binding optimization.** 3D fitting to optimize AutoDock 4.2 grid point spacing and grid box Z-center for ARB binding to the apo-$AT_1R$ model. The fit providing the optimal parameters is listed below the ARB. The seven-by-seven table best-modeled Telmisartan; thus, Telmisartan is not shown.
(TIF)

**S4 Fig. Identification of the ligand-binding pocket. A**, Residues lining the binding pocket were identified in the structure, and the distances between them determined to create a search space for water. **B**, The same residues were used to create a breadth-first search of the space identified by the intersection of the ligand-binding pocket centroid sphere and at least one 8 Å sphere from each residue atom listed in the methods. The color scale represents the area within the centroid sphere overlapping with one to seven of the residue spheres, and the purple lines are a backbone trace of the $AT_1R$. The image displayed matches the frame used to create the apo-$AT_1R$ model.
(TIF)

**S1 Text. Script utilized to optimize AutoDock 4.2 parameters.** The script is written in MS Word and cannot be cut and pasted to run due to different handling of fonts. Notes to help the

user are in red boxes on the left-hand side of the text.
(PDF)

**S2 Text. Script utilized to measure the ligand-binding pocket.** The script is written in MS Word and cannot be cut and pasted to run due to different handling of fonts. Notes to help the user are in red boxes on the left-hand side of the text.
(PDF)

**S1 File. apo-AT$_1$R model.**
(PDB)

## Acknowledgments

The authors greatly appreciate Martin C. Michel for providing the data reported in their comprehensive review of ARBs [21], allowing for the creation of Fig 6.

## Author Contributions

**Conceptualization:** Bradley T. Andresen.

**Data curation:** Shane D. Anderson, Asna Tabassum, Wesley M. Botello-Smith.

**Formal analysis:** Shane D. Anderson, Bradley T. Andresen.

**Investigation:** Shane D. Anderson, Asna Tabassum, Jae Kyung Yeon, Garima Sharma, Priscilla Santos, Tik Hang Soong, Yin Win Thu, Isaac Nies, Tomomi Kurita, Andrew Chandler, Rhye-Samuel Kanassatega.

**Project administration:** Bradley T. Andresen.

**Resources:** Shane D. Anderson, Asna Tabassum, Jae Kyung Yeon, Garima Sharma, Priscilla Santos, Tik Hang Soong, Yin Win Thu, Isaac Nies, Andrew Chandler, Rhye-Samuel Kanassatega, Yun L. Luo, Wesley M. Botello-Smith.

**Software:** Abdelaziz Alsamarah, Wesley M. Botello-Smith.

**Supervision:** Yun L. Luo, Wesley M. Botello-Smith.

**Visualization:** Shane D. Anderson, Asna Tabassum, Wesley M. Botello-Smith.

**Writing – original draft:** Shane D. Anderson.

**Writing – review & editing:** Yun L. Luo, Bradley T. Andresen.

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
