## [Decision Letter · Decision Letter 0]

9 Mar 2020

Dear Dr. Andresen,

Thank you very much for submitting your manuscript "In silico prediction of ARB resistance: A first step in creating personalized ARB therapy" for consideration at PLOS Computational Biology.

As with all papers reviewed by the journal, your manuscript was reviewed by members of the editorial board and by several independent reviewers. In light of the reviews (below this email), we would like to invite the resubmission of a significantly-revised version that takes into account the reviewers' comments.

We cannot make any decision about publication until we have seen the revised manuscript and your response to the reviewers' comments. Your revised manuscript is also likely to be sent to reviewers for further evaluation.

Sincerely,

Alexander MacKerell

Associate Editor

PLOS Computational Biology

Daniel Beard

Deputy Editor

PLOS Computational Biology

Reviewer's Responses to Questions

**Comments to the Authors:**

Reviewer #1: In the submitted paper, authors used AT1R and using computer modeling they created more than 100 mutations to the receptor in order to better understand how these mutations affect the binding affinities of known angiotensin receptor blockers. Although the idea is novel, there are lots of serious issues which are needed to be properly addressed:

1) Authors stated that they used 4ZUD pdb coded AT1R and they stated that "Specifically, the N-Terminus (residues 1 to 25), intracellular loop 2 (residues 134 to 140),extracellular loop 2 (residues 186 to 188), intracellular loop 3 (residues 223 to 234), and helix 8 (residues 305 to 316) were added to the AT1R in accordance to the human AT1R sequence and

PDB:4YAY." It is not clear, why authors did not use 4YAY pdb coded AT1R directly? 4YAY also have a selective inhibitor at the binding pocket.

2) Authors mentioned that "Despite the difference in the distances, structural alignment of the DRY, PIF, and NPxxY motifs between the empty AT1R model and inactive A2AR crystal structure (PDB:3EML) demonstrate that the empty AT1R model is inactive (Figure 2). The NPxxY motif is not as clearly aligned as the DRY and PIF, but this is primarily due to the unique position of helix 8 in the empty AT1R model. For further comparison, the empty AT1R model was aligned to

the active mu-opioid receptor (PDB:5C1M), demonstrating the drastic shifts in the DRY, PIF, and NPxxY required to assume the active state." It is not clear why the authors did not use directly active and inactive states from available angiotensin receptors? (i.e. 6do1)

3) It is well known that GPCRs may exist and function as monomers; however, they can assemble to form higher order structures, and as a result of oligomerization, their function and signaling profiles can be altered. Recent findings reveal that AT1R can form homodimers and activate the noncanonical (β-arrestin-mediated) pathway. Thus, effect of dimerization to the binding pocket should not be overlooked.

4) It is stated that "Although there is little experimental data for direct comparison, the

docking correctly predicted the direction of the change in affinity 60% of the time and was within

two times the calculated error 60% of the time" It is not clear how this ratio is found?

5) In Figure 3, instead showing the volume change with single snapshot, authors may consider to show volume change during the simulations.

6) A better representation of the results is needed. Provided details in discussion and results are not enough.

Minor points:

There are 15 authors for this full computational study. It would be good to add a section showing each author's contribution to the paper.

Reviewer #2: The review is uploaded as an attachment.

**Have all data underlying the figures and results presented in the manuscript been provided?**

Reviewer #1: Yes

Reviewer #2: Yes

PLOS authors have the option to publish the peer review history of their article (what does this mean?). If published, this will include your full peer review and any attached files.

Reviewer #1: Yes: Serdar Durdagi

Reviewer #2: No
---

## [Decision Letter · Decision Letter 1]

6 Sep 2020

Dear Dr. Andresen,

We are pleased to inform you that your manuscript 'In silico prediction of ARB resistance: A first step in creating personalized ARB therapy' has been provisionally accepted for publication in PLOS Computational Biology.

Best regards,

Alexander MacKerell

Associate Editor

PLOS Computational Biology

Daniel Beard

Deputy Editor

PLOS Computational Biology

Reviewer's Responses to Questions

**Comments to the Authors:**

Reviewer #2: minor corrections:

page 5 - phosphatidylcholine is a group of phospholipids that has choline as a head group. In the following text you state that simulations carried out in POPC:Cholesterol mixture. Please add correct name of the lipids used.

page 9 - crustal = crystal ?

page 15 - sold benchmark = solid benchmark?

**Have all data underlying the figures and results presented in the manuscript been provided?**

Reviewer #2: Yes

PLOS authors have the option to publish the peer review history of their article (what does this mean?). If published, this will include your full peer review and any attached files.

Reviewer #2: No

---

## [Editor Report · Acceptance letter]

9 Nov 2020

PCOMPBIOL-D-20-00191R1 

In silico prediction of ARB resistance: A first step in creating personalized ARB therapy

Dear Dr Andresen,

I am pleased to inform you that your manuscript has been formally accepted for publication in PLOS Computational Biology. Your manuscript is now with our production department and you will be notified of the publication date in due course.

With kind regards,

Matt Lyles
